# Plasma-Activated Water Promotes Wound Healing by Regulating Inflammatory Responses

Shuai Wang [1,2,†], Dehui Xu [1,2,†], Miao Qi [2], Bing Li [2], Sansan Peng [1], Qiaosong Li [1], Hao Zhang [1,*] and Dingxin Liu [1,2,*]

1  Centre for Plasma Biomedicine, State Key Laboratory of Electrical Insulation and Power Equipment, Xi'an Jiaotong University, Xi'an 710049, China; shuaiwang3059@stu.xjtu.edu.cn (S.W.); dehuixu@mail.xjtu.edu.cn (D.X.); peng33@stu.xjtu.edu.cn (S.P.); liqiaosong0318@163.com (Q.L.)
2  The School of Life Science and Technology, Xi'an Jiaotong University, Xi'an 710049, China; qimiao@stu.xjtu.edu.cn (M.Q.); lb905533242@stu.xjtu.edu.cn (B.L.)
*  Correspondence: zhang216@mail.xjtu.edu.cn (H.Z.); liudingxin@mail.xjtu.edu.cn (D.L.)
†  These authors contributed equally to this work.

**Abstract:** Infection can hinder the process of wound healing, so it is important to begin antibacterial treatment quickly after a wound forms. Plasma activated water (PAW) can inactivate a variety of common wound infection bacteria. In this study, we compared the effects of PAW prepared with portable surface discharge plasma equipment and medical alcohol on wound healing in a mouse full-thickness skin wound model. The effectiveness of wound healing processes in mice was ranked accordingly: PAW treatment group > medical alcohol treatment group > control group. In order to further understand the mechanism of PAW in promoting wound healing, we tested the expression levels of the pro-inflammatory factors interleukin (IL)-1β and IL-6, the anti-inflammatory factor IL-10, and vascular endothelial growth factor (VEGF). The results showed that PAW promoted the release of pro-inflammatory factors and anti-inflammatory factors from the wounds in mice, which allowed the mice in the treatment group to transition out of the inflammatory period early and enter the next stage of wound healing. The expression level of VEGF in the wounds of mice in the PAW treatment group was higher, which indicates that the microvessels around the wound in the PAW treatment group proliferated faster, and thus the wound healed faster. PAW biosafety experiments showed that PAW did not significantly affect the appearance, morphology, or tissue structure of internal organs, or blood biochemical indicators in mice. In general, PAW prepared via portable devices is expected to become more widely used given its convenience, affordability, and lack of side effects in promoting wound healing.

**Keywords:** plasma-activated water; medical alcohol; wound healing; inflammatory factor; vascular endothelial growth factor

## 1. Introduction

The formation of wounds involves a certain degree of randomness and universality. When the skin is damaged, the body immediately initiates the relevant biological and chemical processes to help the skin repair the wounded area. Skin wound healing is not a simple, linear process, but a complex dynamic process involving spatial and temporal variables. It can generally be divided into four overlapping periods: hemostasis, inflammatory reaction, cell proliferation, and tissue remodeling [1–3]. Wounds can generally be categorized as either acute wounds or chronic wounds. Acute wounds are generally caused by external force injuries. Due to the complexity of acute wound formation, the wound is susceptible to infection with a variety of aerobic and anaerobic microorganisms. *Pseudomonas aeruginosa* and *Staphylococcus aureus* play important roles in acute wound bacterial infection, and eventually create chronic wounds that are difficult to heal [4]. Hard-to-heal wounds not only force an economic burden on patients, but also reduce their quality of life. In clinical

wound treatment, traditional antibacterial techniques are often used to intervene in wound healing, such as antibiotics and special wound dressings. The increasing application of antibiotics and wound anti-infection products has contributed to microbial resistance, and often involves side effects, such as slow wound healing. Moreover, traditional treatment methods are expensive and time-consuming, and patients need to seek medical attention for treatment, which is inconvenient. Therefore, there is a need for new methods to promote wound healing, which not only have to be simple to operate, affordable, and free of side effects, but also must be able to handle different types of wounds caused by a variety of injuries in a more timely and convenient manner.

Plasma is the fourth state of matter (after solid, liquid and gas). The macroscopic temperature of non-thermal equilibrium in plasma can be as low as 300 K, which is close to room temperature (non-thermal equilibrium plasma is also called cold plasma). As the generation of cold plasma does not require a special vacuum environment and can occur under atmospheric pressure, it is also referred to as cold atmospheric plasma (CAP). The main chemically active groups in CAP are ions, free electrons, neutral particles, free radicals, and electromagnetic waves (UV, electric field, and visible light) [5–7]. The active species produced by CAP mainly include reactive oxygen species (ROS) and reactive nitrogen species (RNS), including $H_2O_2$, OH, $O_2^-$, $O_3$, $NO_2^-$, $NO_3^-$, NO, and $ONOO^-$ [8–14]. In recent years, due to the high activity levels of CAP particles, they have received much attention in the field of biomedicine, forming a new discipline: plasma medicine. Plasma medicine is a new and innovative field combining plasma physics, life sciences, and clinical medicine. The first application of CAP in the biomedical field was in 1996. Dr. Laroussi used glow discharge plasma at atmospheric pressure to sterilize instruments and food, proving that plasma can effectively inactivate bacteria [15]. In the past 20 years, due to the physical and chemical advantages of atmospheric pressure cold plasma, its applicability in biomedicine has increased, especially in the areas of sterilization, dental diseases, skin care, skin disease treatment, cancer treatment, wound healing, etc. [16–28]. Infection is a major factor that hinders wound healing, so CAP, a new technology that can sterilize and painlessly treat wounds, offers unique advantages in the field of wound healing. Studies have shown that the direct treatment of CAP can not only promote wound healing in terms of wound anti-infection, but it also promotes the proliferation and migration of two key skin cells (keratinocytes and fibroblasts) involved in wound healing, thereby accelerating the healing of the wound [29,30]. In addition, CAP has been shown to accelerate wound healing by producing relevant growth factors, recruiting granulocytes, and promoting angiogenesis around the wound [31–33].

Therefore, CAP has great potential in wound healing, and its use is expected to expand. However, current research mostly uses the direct treatment of CAP to treat wounds, which still has many associated problems. Firstly, the current CAP system is bulky and complicated, which makes it hard to treat wounds early, which can lead to wound infection. Secondly, considering the diversity of the scenes at which a skin wound can occur, it is inevitable that the wound will be exposed to microorganisms in the external environment. In order to prevent infections, it is necessary to disinfect and clean the wound. However, direct plasma treatment cannot clean the wound, and it is difficult to treat the wound uniformly. Therefore, we have designed portable surface discharge plasma equipment ($300 \times 250 \times 300$ mm), which can quickly produce the required PAW to treat skin wounds in time. Timely treatment can also help avoid reductions in the biological effectivity of PAW due to time delays. In our previous research, we proved that the application of the PAW prepared with this portable device for 5 min can inactivate the most common bacteria present in wound infections [34]. In this paper, firstly, some discharge characteristics of the portable plasma equipment, and the concentrations of several important particles in the PAW, are tested. Then, we compare the effects of PAW and medical alcohol on wound healing in a mouse model of full-thickness skin wounds. In order to further understand the mechanism of PAW in promoting wound healing, we tested the related inflammatory factors and vascular endothelial growth factor (VEGF).

When testing new technology, one must consider its security, so lastly we evaluated the biological security of PAW.

## 2. Materials and Methods

### 2.1. Introduction of Plasma Device

This research used portable surface discharge plasma equipment that was independently developed by the Plasma Biomedical Research Center of Xi'an Jiaotong University. It has the advantages of small size and simple operability. An internal connection schematic diagram of the equipment is shown in Figure 1A. The system includes an alternating current (AC) boost system, a direct current (DC) output system, a plasma generation module, an air pump, a cooling fan, and a gas–liquid reaction tank. The plasma generator, with a dielectric barrier structure, is composed of a high-voltage copper plate electrode, a metal mesh ground electrode, and a 1 mm thick piece of polytetrafluoroethylene sandwiched between the two. The high-voltage electrode is connected to the high-voltage output end of the AC output system. The ground electrode has a metal mesh structure composed of multiple hexagons, which is connected to the ground terminal of the AC output system. When the device is supplied with 220 V and each branch switch is closed, surface plasma will be generated on the surface of the mesh ground electrode. The air pump blows ambient air into the gas reaction chamber at a flow rate of about 15 L/min, and blows the generated plasma into the external solution, while simultaneously providing new gas to the gas reaction chamber. A photo of the plasma discharge is shown in the upper right of Figure 1A, demonstrating that the plasma in each hexagonal grid has good uniformity. We used high-voltage probes (Tektronix, P6015A) and current probes (Tektronix, P6021) to measure the voltage and current of the AC boost system, and observed these with an oscilloscope (Agilent, DSO-X 2014A). The result is shown in Figure 1B. The voltage output of the AC boost system was a standard sinusoidal voltage with a peak-to-peak value of 7.5 kV and a frequency of 20 kHz. A physical map of the portable surface discharge air plasma equipment is shown in the lower right of Figure 1A ($300 \times 250 \times 300$ mm). The portable equipment is easy to transport when needed and can quickly prepare the PAW for treating wounds.

### 2.2. Emission Spectrum Detection

A UV-Vis spectrometer (Maya pro 2000, Ocean Optics, Shanghai, China) was used to measure the surface discharge emission spectra. The detection wavelength range was 200~800 nm. We placed the spectrometer probe about 2 cm away from the surface discharge area to ensure the clarity and accuracy of spectrum detection.

### 2.3. Detection of ROS and RNS in PAW

Here, we used chemical fluorescent probes to measure the concentrations of long-lived species in the PAW. The concentration of $H_2O_2$ in the PAW was measured with a hydrogen peroxide assay kit (Beyotime, Nantong, China). Additionally, a total nitric oxide assay kit (Beyotime, Nantong, China) was used for $NO_2^-$ and $NO_3^-$ measurement.

In this study, electron spin resonance (ESR) spectroscopy was used to detect the concentrations of $ONOO^-$ and $O_2^-$, two short-lived particles that play key roles in biological processes. The lifespans of these two kinds of particles are very short, and it is difficult to directly measure their actual concentration. Therefore, in practical applications, a spin trap is needed to capture the concentration of the particles. TEMPONE-H (1-hydroxy-2,2,6,6-tetramethyl-4-oxo-piper-idine; 1mM; Enzo) is the trapping agent used for detecting the concentration of $O_2^-$, and it can generate the stable adduct TEMPONE. However, TEMPONE-H can also react with $ONOO^-$ to generate TEMPONE, so we used superoxide dismutase (SOD; 100 μg/mL; Sigma) extracted from bovine red blood cells to remove $O_2^-$. Thus, the final concentrations of $O_2^-$ and $ONOO^-$ in the liquid phase can be calculated. TEMPONE-H and SOD were added to the solution before plasma treatment.

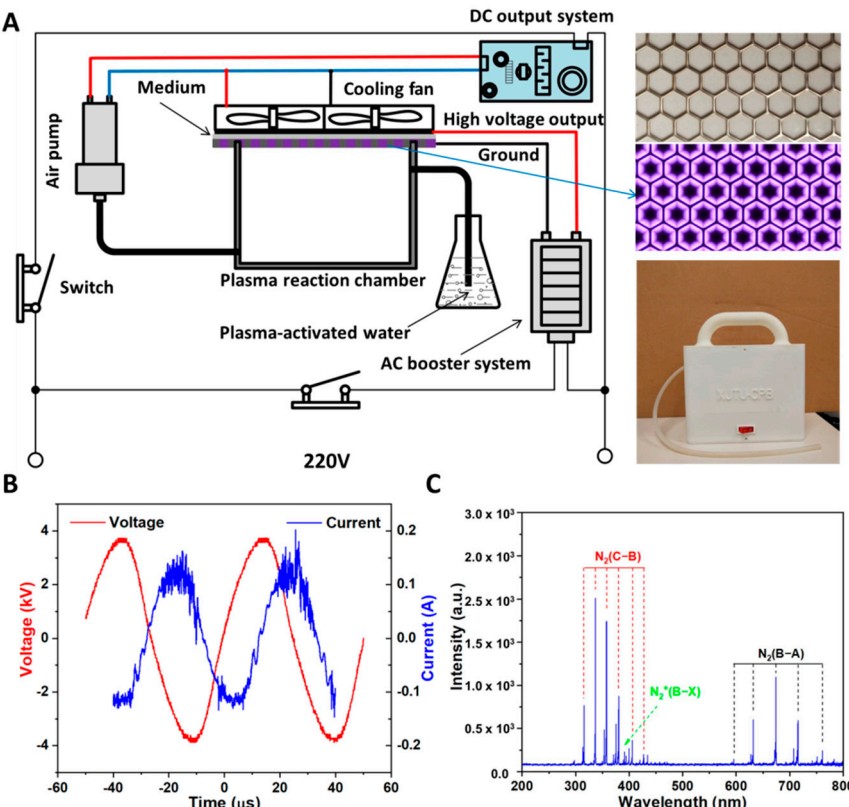

**Figure 1.** Experimental device and its electrical characteristics. (**A**) Internal schematic diagram, physical map, and discharge photo of the portable plasma device. (**B**) Waveforms of applied voltage and discharge current. (**C**) Emission spectrum of the surface discharge plasma.

### 2.4. Animal and Wound Model Creation

In this experiment, we purchased male Institute of Cancer Research (ICR) mice (specific pathogen free, $30 \pm 5$ g) from the Animal Center of Xi'an Jiaotong University to use as our research objects and raised them in the standard breeding environment of the Animal Experiment Center of Xi'an Jiaotong University. In this study, the design and operation of all experiments were carried out in accordance with animal welfare guidelines and were approved by the ethics committee (approval no.: xjtu2018-215). When the mouse full-thickness skin wound model was created, each mouse was injected with 1% sodium pentobarbital at a dose of 0.008 mL/g according to its body weight. After the mice became unconscious, we used a shaving device and depilatory cream to remove their back hair and wiped the depilated areas with medical alcohol. Then, we used sterilized scissors to cut a circular wound with a diameter of 2 cm into the hairless area of each mouse and cut a layer of muscle membrane under the skin to facilitate the construction of an infected wound model. Then, 200 μL ($5 \times 10^6$/mL) of *Pseudomonas aeruginosa* was dropped onto the wound, and a piece of sterile gauze was fixed over the wound. After the wound was infected, the mice were randomly divided into three groups (control group, PAW treatment group, and alcohol treatment group) for the subsequent experimental treatments.

### 2.5. Treatment of Wounds

After the wound was infected, we used the portable plasma device described above and activated 100 mL of tap water for 5 min to prepare the PAW for the experiment. For the skin wound of each mouse in the PAW treatment group, we used a syringe to draw 10 mL of PAW and rinse at a uniform speed. Each mouse in the control group was treated with an equal amount of tap water, and mice in the alcohol treatment group were gently

wiped with a cotton ball soaked in medical alcohol. We performed the same treatment on days 0, 1, 2, 4, and 8.

### 2.6. Analysis of Wound Healing

Throughout the PAW treatment of wounds, we recorded the wound healing of the mice in the control, PAW treatment, and alcohol treatment groups through photos taken at different time points. When taking pictures, we placed a ruler alongside the wound to measure the changes in wound size in mice from different groups. Additionally, the camera was placed at the same height each time. The time (in days) from wound treatment to wound healing in the PAW treatment, control, and alcohol treatment groups was recorded. In this experiment, all wound measurements were completed by the same experimenter to avoid uncertainty errors.

### 2.7. Detection of Inflammatory Factors

We took skin from the wounds of mice from the control group and PAW-treated group on days 1, 2, 3, and 5, and stored them at $-80\ ^\circ$C prior to analysis. In this experiment, an enzyme-linked immunosorbent assay (ELISA) was used to detect inflammatory factors in the wounded skin samples. The protein levels of three inflammatory factors, IL-1$\beta$, IL-6, and IL-10, in the skin samples were measured separately using a mouse IL-1$\beta$ uncoated ELISA kit (Catalog Number 88-7013, Invitrogen, Carlsbad, CA, USA), a mouse IL-6 uncoated ELISA kit (Catalog Number 88-7064, Invitrogen, USA), and a mouse IL-10 ELISA kit (EK210/4-48, MultiSciences, Shanghai, China) according to their instructions.

### 2.8. Detection of Vascular Endothelial Growth Factor

On the days 5 and 8, skin samples from mice in the control and PAW-treated groups were placed in 4% paraformaldehyde for fixation. The samples were sent to Wuhan Seville Biotechnology Company (Wuhan, China) to detect the expression of VEGF in the samples via immunohistochemistry.

### 2.9. Biosafety Analysis of PAW

At the end of the mouse skin wound healing experiment, the control and PAW-treated group mice were first anesthetized, and then blood was collected from the mice by cardiac puncture. The blood was kept in a 1.5 mL centrifuge tube at room temperature for 2 h. It was then centrifuged at 3000 rpm for 15 min, and the supernatant was assessed for blood biochemical indicators. The mice were sacrificed after the blood samples were taken and were then fixed on the operating table for the dissection and separation of the internal organs (heart, liver, spleen, lung, and kidney). First, we gently washed blood stains from the surfaces of the organs in normal saline, and then used a paper towel to absorb the normal saline on the surface. Each organ was photographed, and then tissue-fixed with 4% paraformaldehyde for 48 h. Serum samples and visceral tissue samples were sent to Wuhan Seville Biotechnology Co., Ltd. (Wuhan, China) for blood biochemical index detection and hematoxylin–eosin staining, respectively.

### 2.10. Statistical Analysis of Data

The data from this study were processed using GraphPad Prism 5.01 (GraphPad Software, San Diego, CA, USA) statistical software. All experimental results are presented as the mean $\pm$ standard deviation (SD) of at least three independent experiments. Student's t-test was used to evaluate statistical significance. "**" indicates $p < 0.01$; "*" indicates $p < 0.05$. $p$ values $< 0.05$ between two independent groups were considered to be statistically significant.

## 3. Results

### 3.1. Plasma Discharge Parameters and Characteristics

In this experiment, surface discharge plasma was generated in ambient air via sinusoidal voltage with a peak-to-peak value of 7.5 kV and a frequency of 20 kHz. Figure 1B shows the sinusoidal voltage waveform and current waveform when plasma is generated. Figure 1C shows the radiation species produced by the surface discharge plasma. Figure 1C also shows that the surface discharge plasma has many radiation lines, such as $N_2$(C-B), $N_2$(B-A), $N_2^+$(B-X), etc.

### 3.2. ROS and RNS Detection in PAW

The active particles in the plasma gas phase dissolve, and a variety of complex chemical reactions occur to form PAW rich in ROS and RNS. Here, we tested some of the more common long-lived particles, such as $H_2O_2$, $NO_2^-$ and $NO_3^-$, in the PAW produced with the portable surface discharge plasma device, as well as some short-lived particles such as $O_2^-$ and $ONOO^-$. The concentration changes that occur in the long-lived particles and short-lived particles in PAW under 3 and 5 min activation are shown in Figure 2A,B. It can be seen from the results that for both long-lived and short-lived particles, the concentration under 5 min activation is higher than that under 3 min activation. Therefore, when applying PAW, we can appropriately increase the activation time to improve its effects.

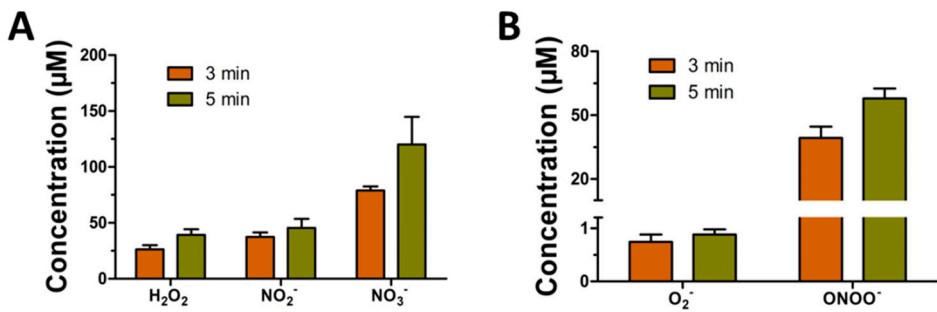

**Figure 2.** Detection of reactive oxygen species (ROS) and reactive nitrogen species (RNS) in plasma-activated water (PAW). (**A**) The concentration of long-lived particles $H_2O_2$, $NO_2^-$ and $NO_3^-$ in the solution after plasma treatment for 3 and 5 min. (**B**) The concentration of short-lived particles $O_2^-$ and $ONOO^-$ in the solution after plasma treatment for 3 and 5 min.

### 3.3. Skin Wound Healing in Mice

We constructed three mouse skin wound infection models, each with 10 mice. The three groups were the control group, the alcohol treatment group, and the PAW treatment group. During wound treatment, the three groups of mice were kept in the same environment and provided with sufficient mouse food and water. The results of wound healing in the three groups of mice over time are shown in Figure 3. In order to evaluate wound healing, we took pictures of the wounds of the mice in the control, alcohol treatment, and PAW treatment groups at different time points (days 0, 2, 4, 8 and 12) (Figure 3A). In order to more clearly assess the wound healing of the three groups of mice, Figure 3B shows results from the statistical analysis of wound diameters on days 0, 2, 4, 8 and 12. Similarly, a histogram of the statistical results of the time required for complete wound healing in the three groups of mice is also shown in Figure 3C. It can be seen from the photo in Figure 3A that the rate of wound healing in the alcohol treatment group was higher than that in the control group, and the wound was smaller than those in the control group on the 12th day. The PAW treatment group had the fastest wound healing rate, and on the 12th day, the wound was significantly smaller than that in the alcohol treatment group and the control group. Figure 3B shows the change in trend for the wound diameter in each group over time. With the passage of time, the wound diameter in the PAW treatment group decreased the fastest, followed by the alcohol treatment group, and then the control group.

This shows that the effect of PAW treatment on wounds is better than traditional alcohol treatment. In addition, we also measured the time from wound modeling to wound healing in each group (Figure 3C). The results showed that the time required for wound healing in the PAW treatment group was the shortest and was significantly lower than that of both the control group ($p < 0.01$) and the alcohol treatment group ($p < 0.05$). Therefore, we conclude that the effectivity of treatments for wound healing in mice in this model can be ranked as follows: PAW treatment group > medical alcohol treatment group > control group.

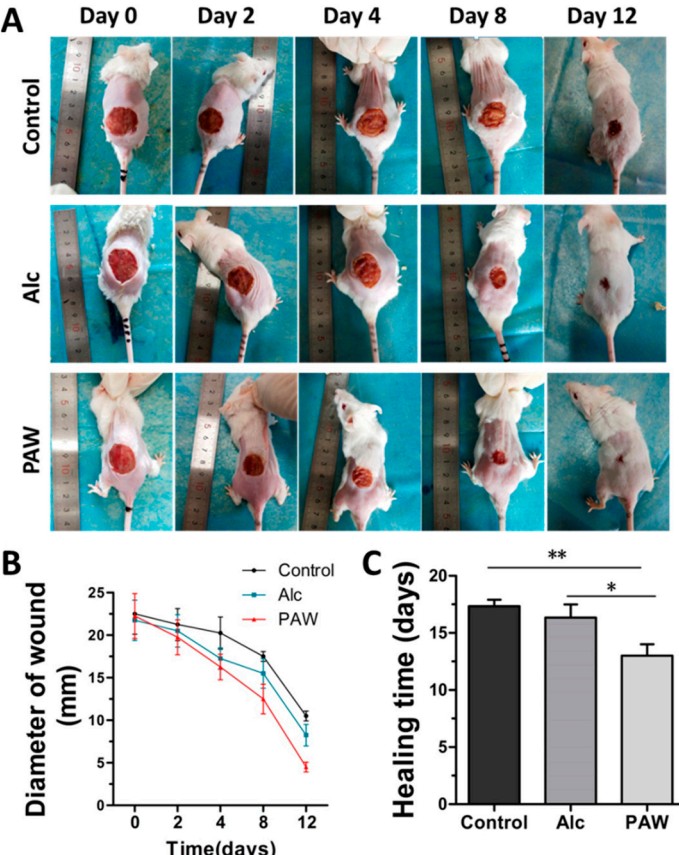

**Figure 3.** Skin wound healing process in mice. (**A**) Photographs of the wounds taken at different time points (days 0, 2, 4, 8, and 12) for the control, alcohol treatment, and PAW treatment groups. (**B**) The results of the statistical analysis of wound diameters in the control group (black line), the alcohol treatment group (blue line) and the PAW treatment group (red line) on days 0, 2, 4, 8, and 12. (**C**) The results of the analysis of the time required for the wounds to completely heal in the three groups of mice. Data are expressed as the mean ± SD, $n = 10$; * $p < 0.05$, ** $p < 0.01$ (Student's *t*-test).

### 3.4. Pro-Inflammatory Factors and Anti-Inflammatory Factors

In this experiment, we used ELISA to detect IL-1β and IL-6, two common pro-inflammatory cytokines that can be secreted and synthesized by keratinocytes, fibroblasts, macrophages, and other cells. Figure 4A,B compares the protein expression of pro-inflammatory factors IL-1β and IL-6 in the wounded skin of mice in the control and PAW treatment groups. The protein expression of IL-1β and IL-6 in the treatment group was significantly higher than in the control group on the first day after PAW treatment. The protein expression of IL-1β and IL-6 in the treatment group began to decrease on days 2, 3, and 5 after PAW treatment, and were lower on the 5th day. The protein expression of IL-1β and IL-6 in the control group mice showed an overall upward trend on days 2, 3, and 5, and were also higher on day 5. In this experiment, we also used ELISA to detect the expression level of the cytokine IL-10. IL-10 is a multi-cell-derived cytokine, which plays a key role in anti-inflammatory processes. Figure 4C compares the anti-inflammatory

factor IL-10 protein expression in the control and PAW treatment groups. Overall, the IL-10 protein content of the control group increased on days 1, 2, 3, and 5, but the IL-10 protein content of the treatment group was significantly higher than that of the control group on the first and second days after PAW treatment. On days 3 and 5, the IL-10 protein content of the PAW treatment group was not as high as that of the control group, but it was also at a high level.

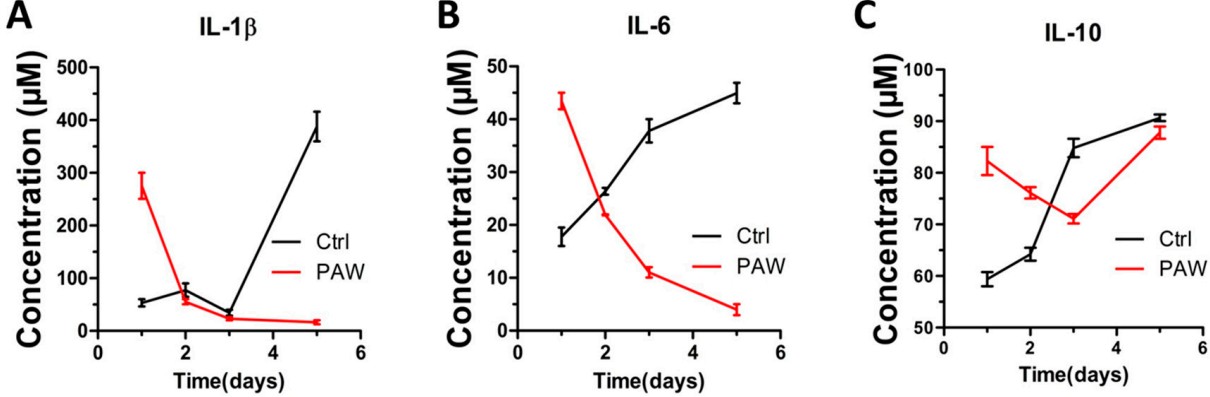

**Figure 4.** Skin inflammatory factor levels. (**A**,**B**) Comparison of pro-inflammatory cytokine expression in the wounded skin of mice in the control and PAW treatment groups at different time points (days 1, 2, 3, and 5). (**C**) Comparison of anti-inflammatory cytokine expression in wounded skin of mice in the control and PAW treatment groups at different time points (days 1, 2, 3, and 5).

### 3.5. Vascular Endothelial Growth Factor

We took wounded skin samples from mice in the PAW treatment and control groups on days 5 and 8 and analyzed the expression levels of VEGF via immunohistochemistry. Figure 5 shows a 100-fold magnified image of the skin sample after immunohistochemical staining, wherein positive expression is colored brown/yellow. The immunohistochemistry analysis shows that the expression of VEGF in the PAW treatment group was higher than in the control group on the fifth day. On the eighth day, the expression of VEGF in the control group increased, but the PAW treatment group had more positive VEGF expression. Assessing immunohistochemistry with the naked eye is a qualitative approach and involves a certain degree of subjectivity. In order to quantify these results, we used ImageJ software to determine the cumulative value of the positive VEGF-expressing pixels in the immunohistochemical photos; that is, the integrated optical density (IOD). The IOD results (in the form of quantitative statistics) are shown in Figure 5. From the figure, we can see that on both days 5 and 8, the IOD value of the positive expression of VEGF in the PAW treatment group is significantly higher than in the control group, and there is a significant difference between the two ($p < 0.001$). On the fifth day, the IOD of the positive expression of VEGF in the control group was $3839 \pm 185$, and $7611 \pm 680$ in the treatment group. On the eighth day, the IOD of the positive expression of VEGF in the control group was $7175 \pm 488$, and $13,218 \pm 673$ in the treatment group.

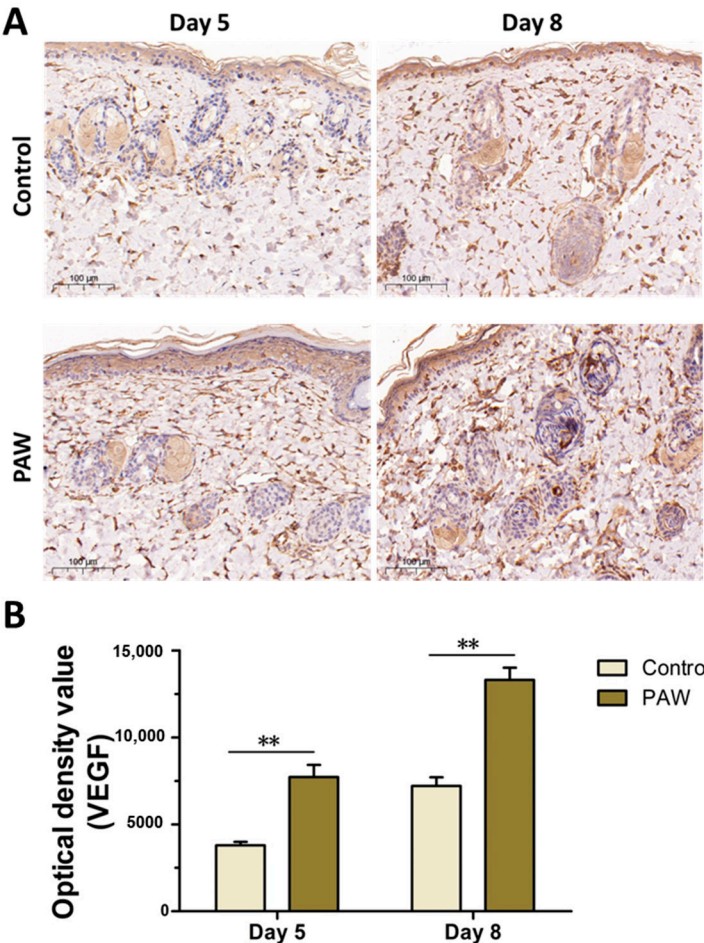

**Figure 5.** Expression level of vascular endothelial growth factor (VEGF). (**A**) Comparison of the expression levels of VEGF in the wounded skin on days 5 and 8 between the control and PAW treatment groups. (**B**) Quantitative statistical results of the positive expression of VEGF. Data are expressed as the mean $\pm$ SD; ** $p < 0.01$ (Student's *t*-test).

### 3.6. Biosafety of PAW

We evaluated the biosafety of PAW by comparing the appearances and morphologies of the main internal organs (heart, liver, spleen, lung, kidney) of the control and PAW treatment groups, and carried out histological analyses of the internal organs and blood biochemical index detection. The results are shown in Figures 6 and 7, and Table 1, respectively. The organs of the three groups of mice displayed no abnormal adhesion during dissection; furthermore, their morphology was regular and complete, their surface was smooth and shiny, and there were no obvious differences between the groups. The results of histological analyses showed that the tissue structures of the main internal organs of the three groups of mice were complete and clear, with no obvious abnormal changes. Table 1 shows that there was no significant difference in the results from the main blood biochemical index of the two groups of mice. This shows that PAW has no significant effect on liver function (albumin and alanine aminotransferase), renal function (urea nitrogen), blood lipids (triglycerides), blood sugar (glucose), organic ions (potassium), or antioxidants (total superoxide dismutase) in mice. Based on the above results, we conclude that PAW is safe for mice.

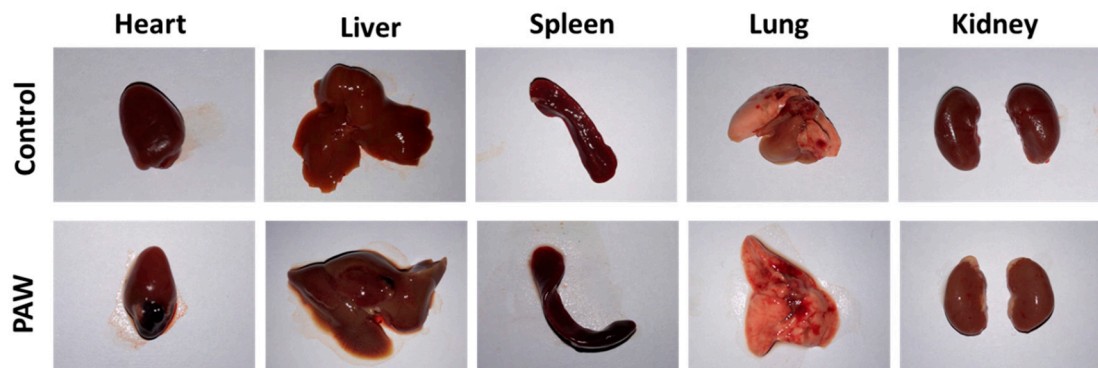

**Figure 6.** The effect of PAW on the external morphology of selected internal organs in mice.

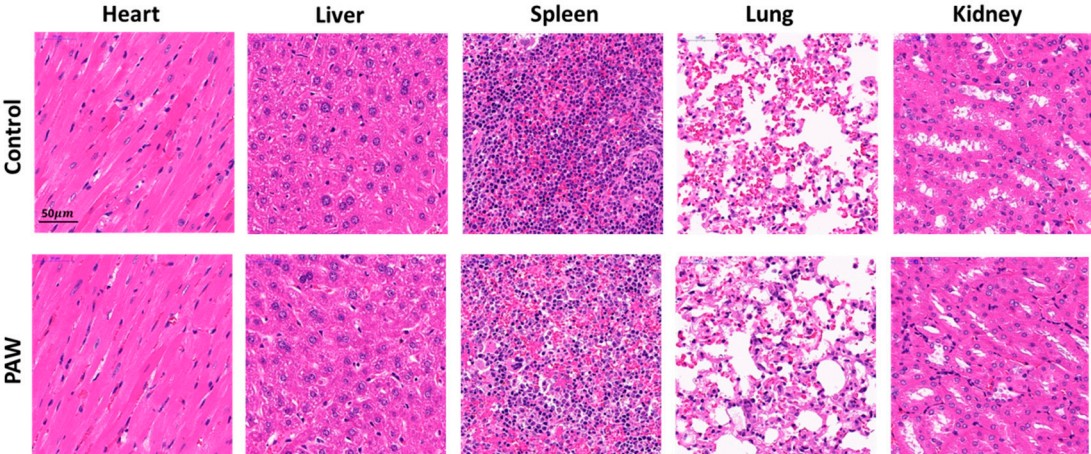

**Figure 7.** Histology of mouse heart, liver, spleen, lung, and kidney stained with hematoxylin and eosin.

**Table 1.** Blood biochemical indicators.

| Indicator | Control | PAW |
|---|---|---|
| Albumin (g/L) | $26.14 \pm 0.43$ | $25.48 \pm 1.05$ |
| Alanine aminotransferase (U/L) | $183.15 \pm 15.40$ | $189.98 \pm 9.85$ |
| Urea nitrogen (mg/dL) | $23.48 \pm 1.52$ | $23.86 \pm 0.46$ |
| Triglycerides (mmol/L) | $2.05 \pm 0.12$ | $2.46 \pm 0.09$ |
| Glucose (mmol/L) | $9.09 \pm 0.57$ | $9.45 \pm 0.92$ |
| Potassium (mmol/L) | $10.85 \pm 0.86$ | $10.67 \pm 1.03$ |
| Total superoxide dismutase (U/mL) | $313.48 \pm 40.91$ | $293.92 \pm 31.2$ |

## 4. Discussion

Relevant research reports indicate that the composition of PAW will change with the increase of storage time, especially the concentration of certain active particles (such as $NO_2^-$) will drop rapidly [35,36]. The PAW equipment used here is a portable plasma equipment, so the equipment can be brought to the use site to prepare the required PAW to reduce the attenuation of active particles in the PAW. Our previous research has proven that PAW can effectively inactivate and reduce the bacterial load of common bacteria in wound infections, thereby promoting wound healing [34]. In order to further illustrate the effect of PAW on wound healing, we have also tested the ROS and RNS that play very important roles in biomedicine [37,38] in the PAW. We found that the PAW prepared by our portable surface plasma equipment is rich in ROS and RNS. This shows that when we use PAW to treat wounds, these active particles in the PAW will stimulate some biological processes related to wound healing. However, it is worth noting that Figure 2 uses a special detection

method (adding a capture agent in advance) to detect the two short-lived particles $O_2{}^-$ and $ONOO^-$ in the solution. The actual survival time of these two short-lived particles is less than one second, so we think these two particles do not have a direct effect on wound healing. Korolov et al. proved through cell experiments that PAW can promote the proliferation of keratinocytes and promote the healing of infected wound models at the cellular level [39]. This shows that PAW can not only promote the proliferation of keratinocytes, but also kill bacteria on infected wounds. In this study, we compared the effects of conventional alcohol and PAW on infected wounds through animal experiments. Medical alcohol has a bactericidal effect and can be used to sterilize medical equipment, skin, etc., as well as killing bacteria in the wound. In order to further explore the effect of PAW on wound healing, we compared the effects of PAW and medical alcohol through a mouse skin infection wound model. According to the wound healing results in Figure 3, we conclude that the PAW treatment group has the best wound healing effect, followed by the medical alcohol treatment group, and the control group was the worst. Therefore, we can conclude that both PAW and medical alcohol can promote wound healing by inactivating bacteria in the wound. However, the effect of PAW in promoting wound healing is more obvious, so we speculate that PAW may promote wound healing through sterilization and in other ways.

Skin wound healing is an extremely complex, dynamic interaction process. When the body is injured, multiple biological pathways in the body are activated at the same time, which initiates complex interactions between cells, between cells and cytokines, or between cells and matrix. The wound healing process involves hemostasis, inflammation, cell proliferation (including connective tissue formation, neovascularization, and skin re-epithelization) and remodeling. The inflammatory reaction period is a very critical stage in the wound healing process. If this stage is extended or inhibited such that it cannot enter the next stage of wound healing smoothly, the wound may become chronic. Here, we examine the levels of related inflammatory factors in the wounded skin of mice at different time points to study whether PAW can accelerate the process of inflammation in the body by adjusting the levels of related cytokines in the inflammatory response phase, thereby accelerating the entire wound healing process. The expression levels of IL-1β and IL-6 are shown in Figure 4A,B. These results indicate that PAW can promote the release of pro-inflammatory factors in mouse wounds, allowing the mice in the treatment group to enter the inflammatory reaction phase of wound healing quickly. Our results are consistent with Schmidt's report (after plasma treatment, the secretion of IL-1β and IL-6 in the wound is stronger and faster) [40]. There is a dynamic balance between inflammatory and anti-inflammatory mechanisms in the inflammatory reaction period. Stimulating the inflammatory response also mobilizes the anti-inflammatory response, such that the body can return to a normal state as soon as possible. Figure 4C shows the expression level of IL-10, a key anti-inflammatory factor. The results show that PAW promoted the release of anti-inflammatory factors in mice, which allowed the mice in the treatment group to exit the inflammatory period early and enter the next stage of wound healing.

Angiogenesis is a necessary process in wound healing, and adequate vascularization is a necessary condition for the delivery of nutrients and oxygen to the wound site. In this experiment, we analyzed the expression of VEGF in the wounded skin of mice via immunohistochemistry. The principle of immunohistochemical analysis exploits an antigen's ability to specifically bind to the corresponding antibody in order to determine the location and quantity of an antigen in cells or tissues. VEGF, also known as vascular permeability factor (VPF), is a highly specific factor that promotes the proliferation of vascular endothelial cells. It not only promotes the proliferation and growth of vascular endothelial cells but can also promote the formation of new blood vessels in the body and enhance the permeability of blood vessels [41]. In the process of skin wound healing, lower levels of VEGF cause a reduction in blood vessel density, which leads to a reduction in the formation of granulation tissue in the wound, and ultimately delays healing. Sufficient levels of VEGF can synergistically promote wound healing through a variety of mechanisms, including the

formation of blood vessels, the deposition of collagen in the skin, and re-epithelialization. It has been reported that mouse skin wounds after plasma treatment are rich in angiogenic factors and capillaries [40]. As shown in Figure 5, our results are consistent with this result. Figure 5 shows that, compared with the control group, the positive expression of VEGF in the skin of the PAW-treated mice is higher; that is, the microvessels around the wound in the PAW-treated mice proliferate faster, and the wound recovers faster.

Previous experiments have proven that the PAW prepared with the portable surface discharge plasma equipment can promote the healing of skin infection wounds. However, in medicine, while we expect a new technology to have good therapeutic effects in clinical treatment, safety is also a factor we must consider. At the end of the article, we studied the biosafety of PAW. It has been reported that plasma treatment of mouse ear skin wounds did not cause adverse effects on mice [42]. We mainly evaluated the safety of PAW by comparing the appearance of mouse internal organs (heart, liver, spleen, lung and kidney) (Figure 6), the histology of mouse internal organs (Figure 7), and the main blood biochemical indicators (Table 1). According to the results, we did not find significant differences in the appearance and histology of the internal organs of the mice in the treatment group and the mice in the control group. Moreover, there was no statistical difference in the biochemical indicators of the mice in the treatment group and the control group. Therefore, we conclude that PAW is safe for mice.

## 5. Conclusions

In this study, we explored the effect of PAW prepared with portable surface discharge plasma equipment on wound healing by constructing full-thickness skin infection wound models on the backs of mice. We first compared the effects of PAW prepared with portable surface discharge plasma equipment and medical alcohol on wound healing. We proved that PAW has a better effect on promoting wound healing compared with the normal control and alcohol-treated groups. This may be related to the abundant ROS and RNS in PAW. The test results of inflammatory factors showed that PAW promoted the expression of IL-1β and IL-6 in mouse wounds. PAW also promoted the release of mouse anti-inflammatory factor IL-10. This suggests that PAW accelerated the inflammatory response period in mice, allowing the mice to advance to the next stage of wound healing. The results of immunohistochemical analyses proved that the positive expression of VEGF in the skin of PAW-treated mice was higher, which indicates that the microvessels around the wound in PAW-treated mice proliferated faster, and thus the wound recovered faster. The appearance and morphology of the main internal organs of mice, the results of histological analysis, and the main blood biochemical indexes showed that PAW had no obvious side effects on mice. Therefore, PAW can accelerate wound healing by regulating the inflammatory response of the wound without causing obvious side effects. The portable device is easy to carry, simple to operate, and can quickly prepare PAW for wound treatment. Moreover, PAW is affordable, and will not cause pain or side effects, and so it has great application potential for wound treatment in the future.

**Author Contributions:** S.W. and D.X. performed experiments, analyzed the data, and wrote the manuscript; H.Z. and D.L. conceived and supervised the study, revised this manuscript; B.L., M.Q. and S.P. participated in the experiment work; Q.L. provided technical support for experimental equipment. All authors have read and agreed to the published version of the manuscript.

**Funding:** This work was supported by the National Natural Science Foundation of China (Grant No. 51521065 and 51837008), China Postdoctoral Science Foundation (2017M610639), State Key Laboratory of Electrical Insulation and Power Equipment (EIPE19309, EIPE20302) and Special Fund of Shaanxi Postdoctoral Science Foundation (2017BSHTDZZ04).

**Data Availability Statement:** Not applicable.

**Conflicts of Interest:** The authors declare no conflict of interest.

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
