# Peer review of "Plasma-Activated Water Promotes Wound Healing by Regulating Inflammatory Responses"

_biophysica, doi:10.3390/biophysica1030022_

Round 1
Reviewer 1 Report
The authors studied the effects of Plasma activated water (PAW) for the wound healing. They compared the treatment using the PAW prepared from surface discharged plasma generator with the medical alcohol to the wound healing process.
To my knowledge this is one of two studies that study the effect of PAW in wound healing in mice. Moreover the approach from the comparison between the conventional method with the plasma treatment, expression analysis of Vascular Endothelial Growth Factor followed by the biosafety assay of the PAW treatment provides a very useful insight of the indirect plasma application in the medical field.
Overall the author presented the study well, from the concise background, material method and the result and discussion well written to show the effects and the effectiveness of the PAW in the wound healing process. In addition this article is match with the scope of the journal.
Above all consideration, I suggest this manuscript can be accepted for publication as is.
Author Response
Thank you very much for taking the time to review this research during your busy schedule.
Reviewer 2 Report
The authors present a very complete study on plasma-treated water in wound healing. The article should be accepted for publication after a few minor points have been considered in the text:
- why were the wounds inoculated with bacteria? was there a group without bacteria? was wound healing better in infected wounds with plama-treated water compared to non-infected wounds?
- the plasma source is interesting, is there another reference on the discharge from a previous paper?
- the authors have investigated the safety of plasma-treated water by investigating various tissues from mice; to embed the findings in recent research, the publication 10.3390/ijms18040868 should be included to the discussion to reflect the findings; the same applies for inflammatory profiling on cytokines such as IL1b, which was investigated in terms of inflammation in a previous study as well 10.7150/thno.29754
- in the discussion, it should be briefly mentioned that superoxide and ONOO- are short-lived and - albeit they have been measured here - probably do not directly contribute to the wound-healing properties observed with plasma-treated water (it is somewhat mentioned in the Figure 2 legend but should be briefly added to the discussion)
- is the production of h2o2, no2-, and no3- similar if instead of tap water, nacl or ddh2o is exposed to plasma (i.e., is the production of RONS dependent on the type of water used)?
Reviewer 3 Report
The manuscript deals with an important issue, the use of plasma-activated water as wound sterilizer. The authors conducted a very detailed investigation, however the manuscript needs some methodological clarifications.
The effect of the plasma and the plasma-activated phoshpor buffered saline (PBS) on the wound healing process has been investigated in details by Korolov et al. J. Phys. D: Appl. Phys. 49 (2016) 035401. The study concentrated on human keratinocyte cell cultures, infected and non-infected, and showed the effect of the incubation time of plasma treated PBS on the wound closer. Authors may discuss their study in line of these previous results. According to this the manuscript should specify how long the plasma-activated water was applied, did it evaporate, how was exactly applied (sponge, syringe?), in conclusion: how long is estimated that the active species had interacted with the wound.
The second point is the stability of plasma-activated water. It should be clarified how long after the plasma treatment was the plasma-activated water applied. As discussed by Kutasi et al. Plasma Sources Sci. Technol. 28 (2019) 095010 the composition of PAW changes during storage and for example the concentration of NO2- drops fast.
The last point is, it is not clear why authors used tap water. Tap water can contain several compounds that may influence the activation process and the ageing of the PAW. For example in the recent paper it has been shown that small Cu ion (and similarly all Fenton type of ions, such as Fe) content of the water can radically change the after-treatment behaviour of PAW Kutasi et al. Plasma Sources Sci. Technol. 30 (2021) 045015.
Besides, tap water may also contain microorganisms, which would shadow the effect of the PAW components.
